# Virtual Healthcare in Rural and Remote Settings: A Qualitative Study of Canadian Rural Family Physicians’ Experiences during the COVID-19 Pandemic

**DOI:** 10.3390/ijerph192013397

**Published:** 2022-10-17

**Authors:** Nahid Rahimipour Anaraki, Meghraj Mukhopadhyay, Margo Wilson, Yordan Karaivanov, Shabnam Asghari

**Affiliations:** 1Centre for Rural Health Studies, Faculty of Medicine, Memorial University of Newfoundland, St. John’s, NL A1B 3V6, Canada; 2Discipline of Emergency Medicine, Faculty of Medicine, Memorial University of Newfoundland, St. John’s, NL A1B 3V6, Canada; 3Labrador-Grenfell Health, Memorial University of Newfoundland, St. John’s, NL A1B 3V6, Canada

**Keywords:** COVID-19, virtual care, community-based participatory approach

## Abstract

Objective: This paper aims to explore the experiences of rural family physicians using virtual healthcare in their clinical practice during the COVID-19 pandemic in Canada. Design: A community-based participatory approach. Setting: Rural and remote communities in Canada. Participants: Thirteen rural family physicians with at least one year of clinical experience. Results: The data illustrate significant issues associated with virtual healthcare in rural healthcare. The adoption of virtual healthcare has been expressed to pose a harsh polarity; the benefit conferred to rural family physicians with the opportunity to have flexible working hours and work at home while interacting with family members is starkly contrasted with the struggles of insufficient financial support to facilitate setting up virtual healthcare for rural physicians, unreliable technological infrastructure, and inadequate technological resources, which are all exacerbated by the lack of adequate health literacy in rural communities. Results were compiled into five major categories underpinning the lived experiences of rural family physicians: 1—potential trade-off between convenience and quality of care; 2—work–family boundaries; 3—patient-doctor communication; 4—technology as barrier or enabler; 5—increased call duration. Conclusion: The differing trends assessed in the findings illustrate the complications faced in providing virtual healthcare, which resonates with the experiences and views of rural physicians. The findings of this study may guide the development of tailored technologies that adjust for the complexity of administering virtual healthcare in rural communities.

## 1. Strengths and Limitations of This Study

The severity of the impact of transitioning towards virtual healthcare (VHC) on the experiences of healthcare providers in rural communities is largely unknown. This study comprehensively explored the experiences of rural family physicians (RFPs) with VHC during the pandemic.We utilized a community-based participatory approach, which is a meaningful method for calibrating insights, leading to greater comprehension of a given phenomenon.The study was conducted in 2021, one year after the outbreak began; therefore, it did not explore the impact of VHC during different stages (i.e., epidemic, pandemic, and endemic).

## 2. Introduction

The beginning of the coronavirus (COVID-19) has brought significant transitive changes in healthcare institutions worldwide. These changes were largely responsive measures to mitigate and restrict coronavirus transmission. The required immediacy of these modifications resulted in a rapid trend toward virtualization in healthcare service channels worldwide [1,2,3]. This unprecedented trend towards adopting VHC possesses a proclivity for significant complications in the context of rural healthcare.

Wong et al. [2] found an overall spike in worldwide telehealth-related searches, reflecting the demand for VHC in caution of the infectious transmission of COVID-19. The most significant observation was found in Canada, followed closely by the United States [2]. This concurs with Chu et al. [1], who claimed that the number of rural patients who experienced at least one telemedicine visit increased by 26.7% in Canada. However, Chu et al. [1] reported that the rate of telemedicine visits was seen to be steeper with urban populations in contrast to rural populations (220 visits per 1000 urban patients and 147 visits per 1000 rural patients). Furthermore, another study found that the median proportion of in-person visits dropped by 81% over the initial wave of the pandemic [4]. The COVID-19 pandemic has shifted healthcare from traditional in-person delivery channels to virtual mediums in Canada. 

A recent study describes a remote practice situation where physicians and patients reported high degrees of satisfaction in the provision of virtual healthcare (over 90% and 90%, respectively) [5]. The insights coincide with other assessments regarding patients’ satisfaction with VHC, particularly decreased travel time and financial considerations. 

However, since 91% of surveyed healthcare providers reported no experience with virtual visits prior to the pandemic, physicians reported decreased confidence in clinical decision-making and increased time spent on documentation. This can impact the quality of care patients receive through VHC [6]. Kane et al. [7] found that physicians considered the principles of primary care were being compromised by VHC. To further illustrate such complications, a survey conducted by a rural lung transplant clinic in Western Canada on physicians providing VHC depicted dissatisfaction stemming from “missing or incomplete blood work or imaging” [5]. 

Additionally, outdated technology could cause network disruptions that would negatively impact the quality of care and was one of the primary causes of negative clinician experiences with VHC [8]. The limitation of technological networks that facilitate channels to rural communities has also shown to be a factor when considering the differential adoption rates of VHC between rural and urban centers [1]. 

Approximately 20% of the Canadian population resides in rural areas, but only 9.3% of physicians work rurally [9]. According to the Chauhan et al. [10] study, one in seven rural physicians plans to leave practice in rural areas within the next two years. Ng et al. [11] showed that there is less than one physician for 1000 people residing in rural areas, while there are more than two physicians for 1000 people residing in urban areas. According to Rourke [12], although shortage of family physicians and specialists is a crucial issue across Canada, physicians in rural and remote communities are far fewer in number and provide a broad scope of practice. 

RFPs in Canada are already struggling with a shortage of medical staff, a lack of human and medical resources, isolation, workload issues, and a higher rate of burnout, the recent demands on RFPs due to the pandemic have brought an extra pressure on an at-risk group of providers [13]. Despite the drastic shift towards virtualization, minimal research has been conducted to explore the experiences and attitudes of RFPs regarding VHC. Our aim is to examine the impacts of VHC during the COVID-19 pandemic on the clinical practice of RFPs in Canada.

To ensure that RFPs’ perceptions and expertise are engaged in all stages of this study, community-based participatory research (CBPR) was utilized. Various benefits are achieved by employing CBPR, including enhancing the application and relevance of the research data by all involved partners; ensuring the research topic is derived from a major concern of the local economy; assembling stakeholders with diverse backgrounds (skills, knowledge, and expertise) to address complex issues; improving the quality, validity, and practicality of the conducted research by engaging the participants’ local knowledge; enhancing the likelihood of overcoming distrust by communities that have traditionally been the “subjects” of such research, and aiming to improve the well-being and health of relevant communities [14].

## 3. Methods

### 3.1. Community-Based Participatory Research

A CBPR study was conducted by inviting RFPs to provide their expertise and be engaged in the decision-making process of all aspects of this study; the process ranged from developing the interview guide to recruiting participants and collecting and interpreting data. During these various procedures, it was ensured that a consensus was reached between all research team members and stakeholders. According to Israel et al. [15], CBPR plays a critical role in public health. It proposes a partnership approach to research where all partners contribute their respective expertise [14]. As such, CBPR is predicated on bolstering part that equitably involves community members, researchers, and organizational representatives and, therefore, provides a meaningful method for calibrating insights and an informed comprehension of a given phenomenon. Naturally, this facilitates the integration of the garnered knowledge for policy formulations and tangible protocols for the benefit of the community.

### 3.2. Recruitment, Data Collection, and Data Analysis

The participants for this study were RFPs with at least one year of experience working in rural Canada; locum physicians, physicians with a restricted license to practice, and rural physicians practicing for less than one year were excluded. Previous experience with virtual care was not considered an exclusion criterion. For the purpose of this study, virtual care is defined as any available mode of communication between physicians and patients, including video calls, phone calls, and chatting services. Invitations were primally shared through email with the RFPs who attended the Research Capacity Building Programs (RCBP) at Memorial University [16] and then through the Society of Rural Physicians of Canada’s RuralMed listserv [17] to RFPs in Canada. The recruitment procedure continued until no further information was obtained and saturation was reached (after nine interviews). In other words, the procedure was halted once the researcher “[saw] similar instances over and over again.” [18]. However, two further interviews were conducted to search for “groups” that extend upon the diversity of data to ensure that saturation is based on the “widest possible range of data.” [18]. We conducted a semi-structured, in-depth, qualitative 30-min telephone interview to collect data. The interview guide included questions about the positive and negative effects of the COVID-19 pandemic on clinical practice, COVID-19 pandemic policies and regulations, strengths and weaknesses of virtual health care, and contextually relevant needs of Canadian rural healthcare, among other considerations. The sampling strategies included purposive (e.g., gender and years of practice) and snowball sampling. The study followed Braun and Clarke’s six-phase guide to conduct the inductive thematic analysis (TA) on the derived data [19]. Furthermore, the extracted themes and codes were presented and discussed with the advisory team and researchers to establish a consensus. After reaching a final consensus on the extracted themes, a study summary was sent to all participants via email for member checking to enhance the credibility of the results. Generalizability is a critical component of qualitative studies to advance scientific knowledge through the extraction, analysis, and synthesis of findings across several studies for a specified phenomenon under similar parameters [20]. Inferential generalizability, as expressed by Lewis et al. [21], will apply to this study. The researcher encourages transferability by providing comprehensive contextual information and a description of the investigated phenomenon for the reader to examine and assess the tangibility of the findings to alternate contexts [20,22].

### 3.3. Participant Engagement

All correspondence with the participants and research team were conducted through email, video conferencing platforms, and phone. We used a flexible partnership method to provide the community members with various levels of involvement throughout the study; “that it may not mean that everyone is involved in the same way in all issues and activities” [14]. Participants of this study had one or both of the following roles: as a participant in qualitative interviews (i.e., eleven RFPs) or as a member of an advisory team to develop study design, design interview guides, recruit participants, and interpret data (i.e., two RFPs). The advisory team of this study was part of the research construction process; they shaped this research based on the participatory approach and contributed to virtually all aspects of the study. However, due to the time demands and required technical skills, the advisory committee was not directly involved in transcribing and coding the interviews. The study design and research questions were calibrated in close communication between the research team and community members (i.e., the advisory team). The number of questions, word selection, and wording of the interview guide were continually checked with the advisory committee via email and virtual meetings. During the interview, participants were encouraged to suggest further questions or topics for additional questions. After reaching a final consensus, the number of questions increased from three to ten. The advisory team was also involved in participant recruitment by suggesting recruitment strategies and sharing the recruitment letter among colleagues who meet the criteria for this study. Snowball sampling was also utilized, where interviewees assisted in recruiting further participants. 

The primary themes were shared with the advisory team through two different methods: virtual meetings and email. Summarized results were also presented via virtual platforms and email to all participants. The themes were developed and expanded based on open dialogue and consensus and received feedback (e.g., verbally via virtual meetings and written via email) from participants.

## 4. Results

Thirteen RFPs across Canada were recruited via email. The participants consisted of 6 males and 7 females ranging in age from 35 to 65 years old with 5 to 35 years of experience in rural practice. Five categories were identified: 1—potential trade-off between convenience and quality of care; 2—work–family boundaries; 3—patient-doctor communication; 4—technology as barrier or enabler; and 5—increased call duration.

### 4.1. Potential Trade-Off between Convenience and Quality of Care

There are significant difficulties associated with accessing healthcare in rural and remote communities. The participants expressed that complications arise from various infrastructural deficiencies in the respective communities. These challenges include the opportunity cost of travelling long geographical distances, limited access to public transportation, out-of-pocket expenses, and mobility impairments. VHC offers a route to overcome these complications and provide rural residents with convenient and accessible healthcare services.

“I think, for many in the rural area…, it’s nice to save a whole ton of time travelling and some of them work in a different community than my office might be in, and that’s sort of been a good thing. I think for me too… because sometimes virtual care is faster. So, instead of having a 3–4 week wait time to come in, I can see people within a few days, usually when they want an appointment.”

However, shifting from in-person visits to virtual care may cause a decline in the quality of care. Some specific diagnoses and clinical decisions cannot be made without in-person medical examinations. For instance, it is difficult to detect the reasons for dizziness or skin rashes over a phone call self-assessment. 

“The big obvious one is rashes. It’s very difficult to assess a rash over the phone… but you know, there’s also saying that…, the patients with asthma want their puffers refilled- it’s easy to ask some questions, but some people don’t realize that they do still have a bit of wheezing in the bottom of their lungs…”

“So, we’re going to miss cancers. We’re going to miss diagnoses that we otherwise would make, and there’s going to be a backlog of things that show up later when we start having in-person visits.”

### 4.2. Work–Family Boundaries

Virtual care has brought numerous benefits along with steep challenges for RFPs. As noted by participants, the opportunity to work at home while interacting with family members is highly valuable. Additionally, working at home provides physicians with a flexible work schedule compared to working in clinics, which comes with the perceived cost of low autonomy and diminished control over working hours.

“The move towards virtual care and for me, it has made all the difference in, like, my life and my practice and things because it has allowed me to be able to come to live near my family.”

Yet, it is challenging to establish and maintain a healthy work-life balance and boundaries while struggling with excessive burdens. These additional demands were assessed to be familial, resulting from additional responsibilities in running household errands and providing childcare (due to the closure of schools during COVID-19); deficiencies in staff communication resulting in reduced support and additional work demands; and erratically long hours of work to meet professional obligations. Additionally, a long call with patients over the phone caused long work hours in an isolated workplace. Participants felt that phone calls were less defined than traditional in-person appointments and added a time cost to practitioners. The interweaving of family commitments and professional obligations has increased the frequency of experienced work-family conflicts, which has been indicated to foster a negative impact on the physical and mental well-being of physicians. 

“That’s a positive for the patient, but that’s kind of a negative to us in terms of burnout [and] sustainability in the system we’re in, which is silo-ed and not supported from an interdisciplinary and collaborative hybrid system. So, that virtual care comes with quite the burden. You often feel you can be always available trying to meet whatever, you know- those needs. So, the long days… the downside that comes with virtual care and the burden of not being always available and I guess… just the whole stress, burnout, and how that factor comes into practice about the added load to the practice”

### 4.3. Patient-Doctor Communication

Resolving patients’ health issues in remote areas and providing them with medical guidance and support to monitor their health conditions is beneficial for patients and physicians. However, poor patient-doctor communication in a virtual setting poses a large set of challenges: dealing with new patients, accessing medical records, difficulties with assessing the understanding and health literacy of patients, the absence of visual cues (e.g., body language and facial expressions), and third-party participation. Dealing with new patients, building relationships over the phone and lack of access to the patient medical records (especially those with no family doctors) are barriers to having a high-quality relationship with patients in virtual care. The low health literacy of patients can lead to miscommunication and misunderstanding, which is intensified by a lack of non-verbal communication. According to physicians, some patients struggle with pronouncing the names of their medications and following medical advice over the phone. 

“They’re trying to tell me like what medications they’re taking, and they, you know, they can’t pronounce the names of the medications.”

The inability to detect facial expressions and gestures of patients during virtual care causes significant differences in patient-physician interactions between in-person and virtual care. For instance, according to physicians, during in-person meetings, it was less challenging to identify signs of misunderstanding; therefore, there were opportunities to mitigate the effects of low health literacy and establish clear communication.

“So, I mean, I think the lack of connection- I mean, for me, one of the things was [that] it was a fairly new practice. So, some people I’ve never met face to face still, and I’ve only, you know, talked to them virtually. So, that’s- that’s certainly a negative, and sometimes you don’t get the nuances of a situation when dealing with things over the phone.”

The value in the supportive role of third-party participation (i.e., a patient support person on the same phone call) for patients with special needs or simply serving as a second set of ears to recall the information provided in virtual care is undeniable. However, miscommunication with third parties who do not always have adequate familiarity with the patients’ health issues not only creates an extra barrier for effective virtual patient–provider communication but also raises potential ethical concerns regarding privacy. 

“You can talk with this other person because I can’t hear you… but it was difficult to determine if I was talking with the right person and if I had consent.”

### 4.4. Technology as a Barrier or Enabler

Although virtual care employs a combination of video calls, phone calls, and chatting services, it can be ineffective for patients that do not have the required technological devices and struggle with a lack of infrastructure, and/or a lack of technological literacy. Even though virtual care provides patients in remote communities with convenient healthcare services, the socioeconomic status and educational level of patients are important factors in successfully implementing VHC. Physicians often struggle with patients who do not have access to a computer or smartphone and certain patient populations who are not able to use technology and do not have the prerequisite digital skills. Therefore, most patient appointments are limited to telephone-based care.

“One of the reasons why I never went ahead and did sort of- like zoom contacts from home is that the majority of my patients would not have been able to navigate that on their own.”

Telephone-based care also came with significant difficulties. Audio calls with certain groups of patients are often cut out due to poor cellphone signals, limited calling plans, or unreliable Internet and cell-service connectivity. For patients with unlimited calling plans, unreliable networks and low bandwidth in rural communities impacted the quality of provider-patient virtual care experiences. This causes additional work for physicians who would have to reschedule unsuccessful appointments. 

“I think the other thing is… that there’s a lot of people [who] that added their Internet or their phone connection; like, their cell service is just not good where they live, and so, we’re asking them to do that, but then at the same time they don’t have the resources or some just financially don’t have- like, they run out of minutes on their phone or things like that.”

Physicians’ most challenging factors are a lack of technological devices and infrastructure. They struggle with the lack of funding and resources to support the necessary technological equipment along with secure firewalls to protect patient information; this leads the respective physicians to deliver virtual care in clinics or, at an early stage of the pandemic, work with their technological equipment at home where there is no support staff. Physicians who managed to transfer technological equipment outside their clinics not only struggled with out-of-date and inadequate technological devices, which put a financial burden on them to obtain the necessary devices, but also reported technical issues related to Internet access and secure firewalls.

“Oh do you want to provide the care from home like you know, do your virtual care from home?” and I said no, I won’t. It’s too complicated to set-up computers, and I live in a rural area, and my Internet’s not that reliable. I’d have to have somebody come in, and you know, because it’ll be firewalls, to protect patient confidentiality; all that kind of stuff. So I said no, I’ll come into the clinic. There’s no problem with that, but I went into the clinic, and there was hardly had any support stuff.”

### 4.5. Increased Call Duration

Although phone calls, as a primary point of contact with patients, are considered a more efficient and convenient means of communication, many phone calls and long telephone conversations overburden and stress physicians. Some physicians believe that phone calls can be complicated due to insufficient medical literacy, technical literacy, infrastructure, and poor provider-patient connections. Alternatively, simply phone calls can result in chatting with patients instead of a clinical visit. 

“Patients don’t treat a phone call from their doctor the same way they treat an in-clinic visit. So, they were often had long chats on the phone. It’s kind of like we’re having a chat instead of a clinical visit, and I found they took a lot longer. There was no way I could finish a patient appointment in less than 20 min.”

Providing counselling services over the phone during the pandemic is among the reasons for long conversations. The physicians reported that pandemic-related disruptions increased the demand for counselling services. School and university closures have dramatically decreased social interactions among the youth and restricted access to mental health resources provided by schools. During the pandemic, disrupted mental health services and shortages of mental health resources in rural, remote communities directly impacted long telephone appointments. Patients unable to access counselling or psychiatric services have tended to share their emotional distress with physicians over the phone.

“The visits became very, very, very long; some patients- I mean many patients. Of course, mental illness is such a part of family medicine and then, even though many patients with mental illness deteriorated, but also patients who we’ve never seen with anxiety and depression. We were told to fit them in- fit in blood pressure pills but when you got on the phone, they were so stressed in the workplace. A lot of counselling visits became quite longer which makes the backload harder; if it is longer, it’s harder to accommodate more people.”

## 5. Member Checking

Following our analysis, we sent a summary of the study findings to all participants to improve the study’s validity, which provided us with positive feedback and some complementary information. For instance, one of the participants asserted that virtual care “has been moved into the mainstream” and is being hailed as a “saviour for rural medicine.” The participant further elaborates that VHC induces a “rapid increase in infrastructure and comfort with the technology.” However, the alteration of channels of provisioning healthcare cannot be done overnight.

## 6. Discussion

This paper does not aim to disregard the tremendous potential and benefits of VHC in rural medicine but rather to articulate the challenges and difficulties of shifting and switching the system from in-person to virtual in rural and remote communities. The period discussed in this study is “COVID-19 time,” where people’s normalcy was constantly interrupted. Over the course of the study, five categorical underpinning trends emerge. These categories provide evidence to illustrate the impact of VHC on the practice and experiences of rural physicians. 

The findings provide further evidence of the deficiencies of VHC, which have been expressed by other literature [5,23]. However, our study additionally found that the fear and anxiety associated with the transmission of the coronavirus disease deterred patient visits in person. This deterrence potentially prompted greater complexities for rural healthcare institutions in the long term. In analyzing these fears and complexities, along with the comparatively lower healthcare literacy and a higher rate of chronic diseases in rural populations [24], relying on VHC to provide healthcare services has prompted concerns and hesitancy by rural physicians. The study found that RFPs find some medical conditions more challenging to diagnose via VHC.

Furthermore, there is minimal literature on the effects of VHC on work-family boundaries for RFPs. This study found that RFPs face significant difficulty in maintaining work-life boundaries and, in some cases, were subject to additional demands interconnected with their professional obligations.

It has been assessed that VHC incurs deterioration in communication between rural physicians and their patients. Comporting with the findings by Coleman [25], it is found that non-verbal communication plays an important role in doctor-patient interactions. All aspects of non-verbal communication are lost over phone calls and may be diminished during communication via video. In leveraging technology, the issues incurred in non-verbal communication can be increased due to faulty internet connections or transmission delays [25,26]. This study found that the loss of these non-verbal cues has led to miscommunication and miscomprehension, which may be amplified due to the lower levels of health literacy and higher degrees of complexity in medical conditions experienced among rural populations [24]. Physicians reported that patients might not be able to accurately convey information about medications or provide ambiguous medical information, which consequently necessitates their visitation to the clinic/hospital. This is potentially fueled by the health disparity expressed between rural and urban populations, which may contribute to the complexity faced by physicians over virtual communication [24,27]. Patients’ health literacy level is pivotal, particularly in virtual communication. Several tools and tests have been suggested to assess patient health literacy. For instance, a brief, bilingual (English and Spanish), three minutes, 6-item literacy assessment named Newest Vital Sign (NVS) is available at no cost at the Pfizer Clear Health Communication Initiative website [28]. Additionally, health literacy can be assessed by observing the behaviours and characteristics of the patients. Patients with low health literacy levels usually make excuses while filling out forms (e.g., “I don’t have my glasses”), provide incomplete medical history to avoid further questions, show signs of nervousness or confusion, etc. [29]. Future studies should consider context-sensitive tools and tests to address patients’ health literacy in rural and remote communities.

The findings further roots assessments derived from existing literature on the asymmetry of available technological infrastructure between rural and urban populations [24,30]. This asymmetry leads to comparatively lower adoption rates of VHC in rural communities [1]. The absence of adequate technological infrastructure to support VHC in rural and remote communities is undeniable [24,31]. Despite the time and economic convenience provided by VHC, the discrepancy in technological literacy and resources between rural and urban communities prompts the comparatively higher complexities associated with providing virtual care in a rural context. This study found that the technological literacy of patients plays an important role in the quality of VHC. Physicians also struggled with the lack of patients’ access to technological resources or the associated skills to maneuver technological devices. In such frequent occurrences, telephone calls were utilized as an alternative which further constrained communication and prompted further complexities faced by physicians. Furthermore, rural physicians were subject to a lack of funding and resources to support the necessary infrastructure to undertake VHC. This led them to provide virtual care in clinics where the support staff was unavailable. The physicians who were able to transfer the required technological resources to their homes found significant difficulties in setting up the technology.

Although the latest generation of electronic medical records provides a patient portal for communication and record keeping, this study was conducted at an early stage of the pandemic while technological support was not widely available. In addition, the participants reported that due to a lack of technological support for patients and patient technology literacy, phone calls were the most convenient mode of communication. Furthermore, this study found that phone calls, as a primary medium of contact, pose significant stress for physicians. As discussed earlier, rural populations tend to have higher rates of chronic diseases and lower rates of health literacy in comparison with urban populations [24]. This study found that these complexities are emphasized when providing healthcare via phone calls. This led to physicians reporting that phone calls pose significant complications, prompting deviations from their traditional course of providing quality healthcare. The participants reported experiencing extended durations of time in assessing patients due to the reasons above, which increased the frequency of experienced work-family conflicts. A critical finding is that longer conversations were partly attributed to providing supplementary counselling services to patients, for which perhaps a suggestion of continued separate “counselling services” to patients via VHC is appropriate. This need for counselling services was further amplified due to pandemic-related disruptions, a continuation of the pre-existing complications with accessing mental health services in rural communities compared to urban communities [24,32]. Thus, this phenomenon led to an increased burden on the workload of rural physicians.

## 7. Limitations

The findings of the study illustrate the experience of Canadian RFPs with VHC. While the result of this study applies to rural family practice, the involvement of patients and decision-makers could have been beneficial in developing long-term plans and commitment to address the issues. One may suggest that age and number of years of experience may play a role concerning using virtual healthcare in practice; this could include the ability to make a diagnosis via the phone or challenges in using virtual systems due to technology literacy. In this qualitative study using purposeful sampling, we did not find any differences by age, gender, or years of practice among RFPs in providing virtual care. An area of further study may consider whether years in practice or a pre-existing relationship with a patient impacts the quality of virtual care. Irrespective of the different geographical contexts, common experiences of providing and maintaining VHC were derived; however, some hyperlocal factors (e.g., involving family medicine residents in virtual care, particularly if residents work in the site, social/structural and organizational factors related to each community, providing context-sensitive resources for healthcare professionals to address health literacy to improve access and uptake) may not be captured.

Furthermore, the study was conducted in 2021, one year after the outbreak began in Canada. The participants experienced different pandemic phases but may still not be exposed to alternate stages. In progressing to a new normal, future studies could explore the impact of VHC on RFPs during different stages (i.e., epidemic, pandemic, and endemic).

## 8. Conclusions

Despite the increasing accessibility of healthcare services due to the rapid virtualization of healthcare channels during the COVID-19 pandemic (which may improve rural healthcare), the critical necessity of investment in local rural healthcare service capacities is undeniable. While virtual care provides accessible and convenient health services, some significant concerns related to unreliable infrastructure, lack of technological equipment, and lack of health literacy in rural and remote communities should be addressed before moving forward with permanent implementation. The results of this study provide decision-makers with information on the impact of VHC on the experiences of RFPs during the COVID-19 pandemic. Furthermore, the study provides rural healthcare institutions with contextually relevant interventions in adopting virtual care in rural areas based on the experiences and livelihoods of rural physicians.

## Data Availability

All data relevant to the study are available on reasonable request to the corresponding author.

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
