# Peer review of "Virtual Healthcare in Rural and Remote Settings: A Qualitative Study of Canadian Rural Family Physicians’ Experiences during the COVID-19 Pandemic"

_ijerph, 2022, doi:10.3390/ijerph192013397_

Round 1

Reviewer 1 Report

This study aimed to explore the experiences of Canadian rural family physicians in using virtual healthcare in their practice during the COVID-19 pandemic. The researchers used a community-based participatory approach and interviewd eleven rural family physicians. Significant issues associated with virtual healthcare were exposed. Here are my comments and suggestions:

 ·       The aim of the study was to explore the experience of physicians in delivering virtual healthcare. However, this issue and what is already known in this field is almost not discussed in the introduction. I suggest placing greater focus on the central subject matter and how this study adds to existing knowledge.

·       The study was conducted during the COVID-19 pandemic and the results reflect the physicians' experiences during this unique period. The article should place greater focus and emphasis on the COVID-19 period. Or at least make it clear that the period discussed is "COVID-19 time" where people's normal was constantly interrupted. This is mentioned, but more in passing than as a focus. Also, I suggest including in the title: "…during the COVID-19 pandemic".

·        There are two sentences describing the aim of the study in the Abstract (lines 14-15 and lines 29-30). I suggest keeping the first one.

·       Do you have an information regarding the number and basic characteristics of Canadian rural family physicians? This information will help to better understand the generalizability of the study.

·       I suggest explaining in more details in the methods section what kind of Virtual Healthcare are suggested in the rural areas.

·       The participants' age ranged from 35 to 65 years old with 5 to 35 years of experience in rural practice. It is possible that the age and the number of years of experience play an important role with respect of using virtual healthcare in the practice. For example: Does the number of years of experience affect the ability to make a diagnosis via the phone? is there an association between age and technology literacy/delivering video appointment? This issue should be discussed.

Author Response

Reviewer: 1

This study aimed to explore the experiences of Canadian rural family physicians in using virtual healthcare in their practice during the COVID-19 pandemic. The researchers used a community-based participatory approach and interviewed eleven rural family physicians. Significant issues associated with virtual healthcare were exposed. Here are my comments and suggestions:

1-The aim of the study was to explore the experience of physicians in delivering virtual healthcare. However, this issue and what is already known in this field is almost not discussed in the introduction. I suggest placing greater focus on the central subject matter and how this study adds to existing knowledge.

Author response (Please see pages 2-3)

Response:  We have added information about delivering virtual healthcare in rural areas and how this study adds to existing knowledge to the introduction.

A recent study describes a remote practice situation where both physicians and patients reported high degrees of satisfaction in the provision of virtual healthcare (over 90% and 90%, respectively) [15] The insights coincide with other assessments regarding patients’ satisfaction with VHC, particularly, decreased travel time and financial considerations.

However, since 91% of surveyed health care providers reported no experience with virtual visits prior to the pandemic, physicians reported decreased confidence in clinical decision making and increased time spent on documentation (Reeves et al., 2020). This has potential to impact the quality of care patients receive through VHC (Reeves et al., 2020).”[28]. Kane et al. [29] found that physicians considered the principles of primary care were being compromised with VHC. To further illustrate such complications, a survey conducted by a rural lung transplant clinic in Western Canada on physicians providing virtual care depicted dissatisfaction stemming from “missing or incomplete blood work or imaging.” [15].

Additionally, outdated technology could cause network disruptions that would negatively impact the quality of care and was seen to be one of the primary causes of negative clinician experiences with virtual care [30]. The limitation of technological networks that facilitate channels to rural communities is also seen to be a factor when considering the differential rates of adoption of VHC between rural and urban centers [1

Approximately 20% of the Canadians population resides in rural areas, but only 9.3% of physicians work rurally [25]. According to the Chauhan et al. [24] study, one in seven rural physicians plan to leave practicing in rural areas within the next two years. Ng et al. [26] shows that there is less than one physician for 1000 people residing in rural areas, while there are more than 2 physicians for 1000 people residing in urban areas. According to Rourke [27] (p. 322), “although there are important shortages of family physicians and specialists in all areas of Canada, physicians in rural areas are far fewer in number and fulfill a wider range of roles”.

RFPs in Canada are already struggling with shortage of medical staff and human resources, lack of medical resources, isolation, workload issues, and higher rate of burnout, “the current demands on RFPs related to COVID-19 have caused an additional strain on an at-risk group of providers” [23] (p.92). Despite the drastic shift towards virtualization, minimal research has been done to explore the experiences and attitudes of RFPs regarding VHC. Our aim is to explore the impacts of VHC during the COVID-19 pandemic on the clinical practice of RFPs in Canada

To assure that RFPs perceptions and expertise are engaged in all stages of this study, community-based participatory research (CBPR) is utilized. Various benefits are achieved by employing CBPR including: enhancing the application and relevance of the research data by all involved partners; ensuring the research topic is derived from a major concern of the local economy; assembling stakeholders with diverse backgrounds (skills, knowledge and expertise) to address complex issues; improve the quality, validity and practicality of the conducted research by engaging the participants’ local knowledge; enhancing the likelihood of overcoming the distrust by communities that have been traditionally been the “subjects” of such research; and aiming to improve the well-being and health of relevant communities [6].

2-The study was conducted during the COVID-19 pandemic and the results reflect the physicians’ experiences during this unique period. The article should place greater focus and emphasis on the COVID-19 period. Or at least make it clear that the period discussed is “COVID-19 time” where people’s normal was constantly interrupted. This is mentioned, but more in passing than as a focus.

Author response (Please see page 8)

Thank you for the comment. To more accurately reflect the study time, we have added the study time to the title and wherever applicable throughout the paper. As an example the following sentence was added to the discussion:

“The period discussed in this study is “COVID-19 time” where people’s normal was constantly interrupted”.

3- Also, I suggest including in the title: “…during the COVID-19 pandemic”.

Author response (Please see page1)

The title was revised to read: “Virtual Healthcare in Rural and Remote Settings: a qualitative study of Canadian rural family physicians’ experiences during the Covid-19 pandemic”

4- There are two sentences describing the aim of the study in the Abstract (lines 14-15 and lines 29-30). I suggest keeping the first one.

Author response (Please see page1)

Revised accordingly to read: “This paper aims to explore the experiences of rural family physicians using virtual healthcare in their clinical practice during the COVID-19 pandemic in Canada.”

5- Do you have an information regarding the number and basic characteristics of Canadian rural family physicians? This information will help to better understand the generalizability of the study.

Author response (Please see page2-3)

We have added information regarding the number and basic characteristics of Canadian rural family physicians to the paper:

“About 20% of the Canadians population resides in rural areas, but only 9.3% of physicians work rurally (Park, 2007). According to Chauban et. Al’s (2010) study, one rural physician in seven (13% of the respondents), have plan to leave practicing in rural areas whitin the next two years. The result of Ng and colleagues’s study (1997) shows that there is less than one physician for 1000 people reside in rural areas, while there are more than 2 physicians for 1000 people reside in urban areas. According to Rourke (2008, p. 322), “although there are important shortages of family physicians and specialists in all areas of Canada, physicians in rural areas are far fewer in number and fulfill a wider range of roles”.”

6- I suggest explaining in more details in the methods section what kind of Virtual Healthcare are suggested in the rural areas. 

Author response (Please see page 3)

Thank you for the comment. Additional information was added to the method section to clarify the virtual care in this study. The method section now read: “Previous experience with virtual care did not consider as an exclusion criterion. For the purpose of this study, virtual care is defined as any available modes of communication between physicians and patients including video calls, phone calls, chatting services.”

In addition, under the result section, page 7, we described RFPs experiences in employing virtual care to provide care to their patients.

“Although virtual care employs a combination of video calls, phone calls, and chatting services, it can be ineffective for patients who are struggling due to not having the required technological devices, a lack of infrastructure, and/or a lack of technological literacy”.

7- The participants’ age ranged from 35 to 65 years old with 5 to 35 years of experience in rural practice. It is possible that the age and the number of years of experience play an important role with respect of using virtual healthcare in the practice. For example: Does the number of years of experience affect the ability to make a diagnosis via the phone? Is there an association between age and technology literacy/delivering video appointment? This issue should be discussed.

Author response (Please see page 10):

Thank you for the comment. In this qualitative study using purposeful sampling, we did not find any differences by age, gender and years of practice. To address this comment, we added the following information in limitation section.

“One may suggest age and number of years of experiences may play role with respect of using virtual healthcare in the practice, this could include, the ability to make a diagnosis via the phone or challenges in using virtual system due to technology literacy. In this qualitative study using purposeful sampling, we did not find any differences by age, gender and years of practice among RFPs in providing virtual care. An area of further study may consider whether years in practice or a pre-existing relationship with a patient impacts the quality of virtual care.”

Reviewer 2 Report

This qualitative study assessed the experiences of Canadian rural family physicians' with regards to virtual healthcare during 2021, using a community-based participatory approach. The study consisted of a process in which a semi-structured interview was developed and eleven telephone interviews with physicians were conducted and analysed. The importance of the experiences and views of family physicians regarding virtual healthcare in this era where is assimilation was dramatically accelerated is undisputable. However, I have some concerns regarding the article:

1. Although the English language and style are appropriate, the article is difficult to read in terms of length and redundancy. A more concise style would improve it (especially the results section).

2. Introduction- the authors elaborate on the increase in the use of telemedicine throughout the introduction. However, they do not mention any previous research (in the introduction not in the discussion) related to issues raised during the study, such as the quality of care in telemedicine versus in-person visits.  

3.  Methods- the authors applied a community-based participatory approach. While the perspective of physician is addressed, those of patients and the relevant healthcare provider organization/s are absent.  I believe these should be acknowledged when discussing the conclusions of the study.

4.  Methods- The authors state that participants were recruited by emails sent to all relevant physicians. However, they also state that "participants played a pivotal rule in recruiting prospective participants". Such an approach may lead to a non-representative sample of participants.

Author Response

Reviewer: 2

This qualitative study assessed the experiences of Canadian rural family physicians' with regards to virtual healthcare during 2021, using a community-based participatory approach. The study consisted of a process in which a semi-structured interview was developed and eleven telephone interviews with physicians were conducted and analysed. The importance of the experiences and views of family physicians regarding virtual healthcare in this era where is assimilation was dramatically accelerated is undisputable. However, I have some concerns regarding the article:

1-Although the English language and style are appropriate, the article is difficult to read in terms of length and redundancy. A more concise style would improve it (especially the results section).

Author response:

The paper was edited and revised.

  1. Introduction- the authors elaborate on the increase in the use of telemedicine throughout the introduction. However, they do not mention any previous research (in the introduction not in the discussion) related to issues raised during the study, such as the quality of care in telemedicine versus in-person visits.  

Author response (Please see page2-3)

We have added information about delivering virtual healthcare in rural areas including telemedicine and how this study adds to existing knowledge to the introduction

knowledge to the introduction.

A recent study describes a remote practice situation where both physicians and patients reported high degrees of satisfaction in the provision of virtual healthcare (over 90% and 90%, respectively) [15] The insights coincide with other assessments regarding patients’ satisfaction with VHC, particularly, decreased travel time and financial considerations.

However, since 91% of surveyed health care providers reported no experience with virtual visits prior to the pandemic, physicians reported decreased confidence in clinical decision making and increased time spent on documentation (Reeves et al., 2020). This has potential to impact the quality of care patients receive through VHC (Reeves et al., 2020).”[28]. Kane et al. [29] found that physicians considered the principles of primary care were being compromised with VHC. To further illustrate such complications, a survey conducted by a rural lung transplant clinic in Western Canada on physicians providing virtual care depicted dissatisfaction stemming from “missing or incomplete blood work or imaging.” [15].

Additionally, outdated technology could cause network disruptions that would negatively impact the quality of care and was seen to be one of the primary causes of negative clinician experiences with virtual care [30]. The limitation of technological networks that facilitate channels to rural communities is also seen to be a factor when considering the differential rates of adoption of VHC between rural and urban centers [1

Approximately 20% of the Canadians population resides in rural areas, but only 9.3% of physicians work rurally [25]. According to the Chauhan et al. [24] study, one in seven rural physicians plan to leave practicing in rural areas within the next two years. Ng et al. [26] shows that there is less than one physician for 1000 people residing in rural areas, while there are more than 2 physicians for 1000 people residing in urban areas. According to Rourke [27] (p. 322), “although there are important shortages of family physicians and specialists in all areas of Canada, physicians in rural areas are far fewer in number and fulfill a wider range of roles”

RFPs in Canada are already struggling with shortage of medical staff and human resources, lack of medical resources, isolation, workload issues, and higher rate of burnout, “the current demands on RFPs related to COVID-19 have caused an additional strain on an at-risk group of providers” [23] (p.92). Despite the drastic shift towards virtualization, minimal research has been done to explore the experiences and attitudes of RFPs regarding VHC. Our aim is to explore the impacts of VHC during the COVID-19 pandemic on the clinical practice of RFPs in Canada.s

To assure that RFPs perceptions and expertise are engaged in all stages of this study, community-based participatory research (CBPR) is utilized. Various benefits are achieved by employing CBPR including: enhancing the application and relevance of the research data by all involved partners; ensuring the research topic is derived from a major concern of the local economy; assembling stakeholders with diverse backgrounds (skills, knowledge and expertise) to address complex issues; improve the quality, validity and practicality of the conducted research by engaging the participants’ local knowledge; enhancing the likelihood of overcoming the distrust by communities that have been traditionally been the “subjects” of such research; and aiming to improve the well-being and health of relevant communities [6].

  1. Methods- the authors applied a community-based participatory approach. While the perspective of physician is addressed, those of patients and the relevant healthcare provider organization/s are absent.  I believe these should be acknowledged when discussing the conclusions of the study.

Author response (Please see page 9)

We acknowledge the comment. In this community based participatory research, the study target group was rural physicians. Some rural physicians (i.e. advisory team) were involved in incorporating research, creating knowledge, and reflection and others were invited to interview. We added following information to the limitation:

“The findings of the study illustrate the experience of Canadian RFPs with virtual healthcare. While the result of this study is applicable to rural family practice, involvement of patients and decision makers could have been beneficial in developing long term plans and commitment to address the issues.”

  1. Methods- The authors state that participants were recruited by emails sent to all relevant physicians. However, they also state that "participants played a pivotal rule in recruiting prospective participants". Such an approach may lead to a non-representative sample of participants.

Author response (Please see page 3-5)

Thank you for the comments.  The method was revised to better clarify the recruitment strategy in this qualitative study:

“We utilized multiple channels to recruit participants: RCBP and SRPC, advisory team, and participants. Invitations were primally shared through email with the rural family physicians who attended in the Research Capacity Building Programs (RCBP) at Memorial University and then through the Society of Rural Physicians of Canada to all RFPs in Canada. Moreover, the advisory team was also involved in participant recruitment by suggesting recruitment strategies and sharing the recruitment letter among colleagues who meet the criteria for this study. Snowball sampling was also utilized; where interviewees assist in recruiting further participants.

Reviewer 3 Report

Overall, telemedicine was an important solution in many areas of the world during the pandemic.  This research looks from the viewpoint of rural family physicians who face particular challenges with infrastructure and possible low literacy of the patient population.  I applaud the researchers for investigating the challenges faced by the RFP in Canada.

In the introduction, some wording seems awkward such as "cautious" on line 55.  I had to look up the meaning of "coheres" on line 57, I think a more common term could be used to assist the reader with understanding the content.  It seems like references go back and forth between styles which also detracts from readability.  Methods includes a rather long description of CBPR which includes numerous quotes and would be better placed at the end of the introduction, certainly before the actual description of the project (first couple of sentences).  It should be broken in a few paragraphs as well.

Were the participants in multiple provinces (were all provinces represented), representative of the entire rural portion or more Eastern?  Did any or all of the RFP have electronic medical records and/or patient portals as a basis for communicating and record keeping? Without mention of this, I feel a loss of context as far as what is available to the RFP.

Overall some good points are reiterated in this paper about the challenges with VHC.  I did find the concern about duration interesting, when the visits had to be on the phone.  From the physician standpoint, this had to be especially difficult to control.

The concept of "member checking" needs more explanation.  Was this just further conversations with the researchers? I assume the members are the RFP?

Going back to the frustration voiced by RFP about lengthy phone calls, perhaps the concept of work-family balance for RFP can be further explained in terms of long work hours in an isolated work place. 

I wondered why 353 "presence of inadequate" was used vs "absence of adequate"? If the former is preferred, I think the sentence can start with "inadequate technological ..."

The paragraph starting with 367 should be tied back with the work life balance as well. Perhaps a suggestion of continued separate "counseling services" via VHC is appropriate (this is common among college campuses).

Under Limitations, "hyperlocal factors" is mentioned, can you provide examples?

Line 397 has "lake" instead of lack. How would you suggesting addressing patient health literacy in the populations included in the study?  Is there anything the RFP or the community can do to assist (tools)?

Author Response

Reviewer:3

Overall, telemedicine was an important solution in many areas of the world during the pandemic.  This research looks from the viewpoint of rural family physicians who face particular challenges with infrastructure and possible low literacy of the patient population.  I applaud the researchers for investigating the challenges faced by the RFP in Canada.

1-In the introduction, some wording seems awkward such as "cautious" on line 55.  I had to look up the meaning of "coheres" on line 57, I think a more common term could be used to assist the reader with understanding the content. 

Author response:

The paper has been edited and revised to increase readability.

2- It seems like references go back and forth between styles which also detracts from readability. 

Author response:

The reference style has been revised.

3- Methods includes a rather long description of CBPR which includes numerous quotes and would be better placed at the end of the introduction, certainly before the actual description of the project (first couple of sentences).  It should be broken in a few paragraphs as well.

Author response (Please see page 3):

Thank you for the comment. The quotes were removed, and some part of description moved to the introduction section.

“To assure that RFPs perceptions and expertise are engaged in all stages of this study, community-based participatory research (CBPR) is utilized. Various benefits are achieved by employing CBPR including: enhancing the application and relevance of the research data by all involved partners; ensuring the research topic is derived from a major concern of the local economy; assembling stakeholders with diverse backgrounds (skills, knowledge and expertise) to address complex issues; improve the quality, validity and practicality of the conducted research by engaging the participants’ local knowledge; enhancing the likelihood of overcoming the distrust by communities that have been traditionally been the “subjects” of such research; and aiming to improve the well-being and health of relevant communities [6].”

4- Were the participants in multiple provinces (were all provinces represented), representative of the entire rural portion or more Eastern? 

Author response:

We utilized multiple channels to recruit Canadian rural physicians across Canada. One of these strategies was sending the invitations via Society of Rural Physicians of Canada's RuralMed listserv. The recruitment procedure continued until no further information was obtained and saturation was reached.  In this study, saturation was reached after nine interviews. However, two further interviews were conducted to search for “groups” that extend upon the diversity of data to ensure that saturation is based on “widest possible range of data." The location of practice for study participants were Alberta, Ontario, Quebec, New Brunswick, Nova Scotia, Newfoundland, Labrador and Northwest Territories.  We did not find any differences by location of practice.

We revised method to clarify the RFPs recruitment process across Canada (page 4). In addition, we added additional details regarding location of practice to the result section (page 5). 

5- Did any or all of the RFP have electronic medical records and/or patient portals as a basis for communicating and record keeping? Without mention of this, I feel a loss of context as far as what is available to the RFP.

Author response (Please see page 9):

“Although the latest generation of electronic medical records provides a patient portal for communication and record keeping, this study was conducted at early stage of pandemic while the technological support was not vastly available. In addition, the participants reported that due to lack of technological support for patients and patient technology literacy, phone call was the most convenience mode of communication”

6- The concept of "member checking" needs more explanation.  Was this just further conversations with the researchers? I assume the members are the RFP?

Author response (Please see page 4& 8):

Member checking is different from the team consensus.  We first discussed all the themes with the team, including advisory team (two rural family physicians) and researchers, then a summary was sent to the study interviewees.

We revised method and result to better clarify the member checking process:

“Furthermore, the extracted themes and codes were presented and discussed with the advisory team and researchers to establish a consensus. After reaching a final consensus on the extracted themes, to enhance the credibility of the results, a summary of the study was sent to all participants via email for a member checking.”

“Following our analysis, we sent a summary of the study findings to all participants to improve validity of the study, which provided us with not only positive feedbacks, but also some complementary information. For instance, one of the participants asserted that virtual care, “has been moved into the mainstream” and is being hailed as a “savior for rural medicine.” The participant further elaborates that VHC induces a “rapid increase in infrastructure and comfort with the technology.” However, the alteration of channels of provisioning healthcare cannot be done overnight.”

7- Going back to the frustration voiced by RFP about lengthy phone calls, perhaps the concept of work-family balance for RFP can be further explained in terms of long work hours in an isolated work place. 

Author response (Please see page 5-6):

Response: thank you for the comments. We have added additional information to further explain the long work hours in an isolated work place under work-family balance.

“Additionally, a long duration of call with patients over the phone, caused long work hours in an isolated workplace. Participants felt that phone calls were less defined than traditional in person appointments and added a time cost to practitioners.”

8- I wondered why 353 "presence of inadequate" was used vs "absence of adequate"? If the former is preferred, I think the sentence can start with "inadequate technological ..."

Author response: The sentence was revised to read: Absent of adequate technological infrastructure

9- The paragraph starting with 367 should be tied back with the work life balance as well. Perhaps a suggestion of continued separate "counseling services" via VHC is appropriate (this is common among college campuses).

Author response (Please see page 9):

“A critical finding is that longer conversations were in part attributed to providing supplementary counseling services to patients, which perhaps require a continued separate "counseling services" to patients via VHC.”

10- Under Limitations, "hyperlocal factors" is mentioned, can you provide examples?

Author response (Please see page 10):

“(e.g., involving family medicine residents in virtual care particularly if residents work in the site, social/structural and organizational factors related to each community, providing context-sensitive resources for healthcare professionals to address health literacy to improve access and uptake)”

11- Line 397 has "lake" instead of lack.

Author response: the word revised accordingly.

12- How would you suggesting addressing patient health literacy in the populations included in the study?  Is there anything the RFP or the community can do to assist (tools)?

Author response (Please see page 9):

Thank you for the comment. To better clarify available tools and test to address the health literacy of patient, further information was added to the discussion.

“Patients’ health literacy level plays a pivotal role, particularly in virtual communication. Several tools and tests have been suggested to assess patient health literacy. For instance, a brief, bilingual (English and Spanish), three minutes, 6-items literacy assessment named Newest Vital Sign (NVS), which is available at no cost at the Pfizer Clear Health Communication Initiative website (Osborne, et al, 2007). Additionally, health literacy can be assessed through observing behaviors and characterizes of the patients. Patients with low health literacy level usually make excuses while filling out forms (e.g., “I don’t have my glasses”), provide incomplete medical history to avoid further questions, showing sign of nervousness or confusion, etc. (Cornett, 2009). Future studies should consider context-sensitive tools and test to address health literacy of patients in rural and remote communities.”

Round 2

Reviewer 1 Report

This is a much-improved paper with the additional information provided.